

# A novel long noncoding RNA (lncRNA), LINC02657(LASTR), is a prognostic biomarker associated with immune infiltrates of lung adenocarcinoma based on unsupervised cluster analysis

Fanming Kong[1,*], Xinyu Yang[2,*], Zhichao Lu[2], Zongheng Liu[2], Yang Yang[3] and Ziheng Wang[4,5]

[1] Department of Oncology, First Teaching Hospital of Tianjin University of Traditional Chinese Medicine, Tianjin, China
[2] Research Center of Clinical Medicine, Affiliated Hospital of Nantong University, Medical School of Nantong University, Nantong, China
[3] Department of Trauma Center, Affiliated Hospital of Nantong University, Jiangsu, China
[4] Department of Clinical Bio-bank, Affiliated Hospital of Nantong University, Jiangsu, China
[5] Centre for Precision Medicine Research and Training, Faculty of Health Sciences, University of Macau, Macau SAR, China
[*] These authors contributed equally to this work.

Corresponding authors
Yang Yang,
yangyang286228@ntu.edu.cn
Ziheng Wang,
wang.ziheng@connect.um.edu.mo

## ABSTRACT

Non-small cell lung cancer (NSCLC) has long been the deadliest malignancy worldwide, with adenocarcinoma (AD) being the most common pathological subtype. Here we focused on the value of LASTR in LUAD. Using expression analysis, enrichment analysis, immune cell infraction analysis, we found that the expression level of LASTR was significantly increased in LUAD tissue. Meanwhile, LASTR was significantly associated with differential infiltration of various immune cells. Kaplan-Meier survival analysis showed that LUAD related with a poor prognosis in terms of OS, PFI, and DSS compared with high-expression LASTR. The enrichment analysis showed that LASTR is related to the pathays like PI3K-AKT signaling pathway. Thus, the present findings could be helpful in a better understand of LASTR in LUAD. RT-PCR was used to verify the high expression of LASTR in LUAD tissues, and the apoptosis of LUAD cell lines was promoted by CCK8 and Transwell experiments to verify the ability of LASTR to promote the migration and invasion of lung cancer cells *in vitro*.

## INTRODUCTION

Lung cancer is the major cause of mortality worldwide with nearly 2.1 and 1.8 million new cases and mortalities correspondingly in 2020, which was divided into different histological types (*Vergara-Fernandez, Trejo-Avila & Salgado-Nesme, 2020*). Lung adenocarcinoma, (LUAD), is the primary subtype of lung cancer and has unique histological morphology and mutational heterogeneity among them. This heterogeneity is not limited to tumor epithelial

cells, which encompasses the tumor microenvironment including the cancer-associated fibroblasts (CAFs), vasculature, infiltrating immune cells, and extracellular matrix (ECM). Currently commonly used clinical prognostic markers for LUAD include pathological grading system and tumor lymph node metastasis (TNM) staging, microvascular invasion, tumor necrosis, and invasion of the collecting system. These clinicopathological risk factors are valuable, but not sufficient, in predicting prognosis and assessment in subgroups of patients with LUAD. Smoking, including primary or secondary exposure to the smoke of tobacco, poses a great predisposing factor for lung cancer (*McSorley et al., 2018*). In 2020 global cancer statistics, new confirmed cases accounted for 11.4% of new malignant cases, and death as a result of lung cancer accounted for 18.0% of total cancer mortalities (*Hsia et al., 2018*; *Williams et al., 2019*). The prime reason why the 5-year survival of lung cancer is <15% is that the diagnosis is not timely (*Boshier et al., 2018*; *Malietzis et al., 2016*). Because symptoms of common respiratory diseases are similar, approximately 75% of advanced or metastatic cancers are misdiagnosed and major surgical opportunities are often missed. With the advancement of modern clinical diagnostic approaches, in spite of the improvement in the overall survival rate as well as the quality of life, the survival rate remains unsatisfactory. Recently, pertaining to precision medicine, crucial carcinogenesis-causing genes can be utilized in cancer therapy as therapeutic targets. Molecules mostly reported for adenocarcinoma pathological detection as well as therapeutic targets consist of epidermal growth factor receptor mutations, acanthus microtubule-associated protein-like genes, and anaplastic lymphoma kinase genes. Even though molecularly targeted therapy has depicted promising clinical effects, the treatment of LUAD patients remains challenging as a result of drug resistance. Consequently, it is important to search for effective molecular pathological diagnosis and prognostic predictors of LUAD.

Currently, immunotherapy is utilized as the first-line treatment for advanced LUAD as well as the consolidation management of patients with locally advanced LUAD (*Alvarez-Dominguez et al., 2015*). Unlike targeted therapy and chemotherapy, immunotherapy does not target the cancerous cells themselves, but the tumor microenvironment (TME) is infiltrated by various types of immune cells. Emerging evidence suggests that immune checkpoint inhibitors (ICIs) can bring successful revolution in the management of advanced LUAD individuals by improving TIL function, acting on the TME, and improving patient immune function, thereby prolonging progression-free interval (PFI). However, a great number of patients fail to gain benefit from immunotherapy and this can be attributed to tumor heterogeneity. There are few studies on the link between biomarkers, tumor-infiltrating immune cells, and immune responses. Therefore, the discovery and recognition of new immune-related gene targets in LUAD to enhance prognosis as well as promote the progress of innovative treatment approaches are still urgent issues to be solved. Consequently, finding more abundant and viable LUAD biomarkers will aid clinical diagnosis.

It is generally accepted now that above 75% of the human genome is functional and is involved in encoding large numbers of ncRNAs (PG & AW). Besides protein-coding genes, ncRNAs, especially Long noncoding RNAs (lncRNAs), are considered central regulators of a variety of biological processes. Pertaining to the ENCODE project, the

human genome is estimated to encode more than 28,000 distinct lncRNAs, many of which are still being discovered and unannotated. Although the functionality of lncRNAs remains controversial, the existence of the control of a myriad of biological processes as regulators is not in doubt. LncRNAs have high tissue specificity, efficiency, and stability, and can be used as possible therapeutic targets and diagnostic or prognostic biomarkers (*Jao et al., 2018*; *Yang et al., 2018*; *Wang et al., 2021*). lncRNAs' involvement in the modulation of various physiological processes has been ascertained, such as cell cycle, cell growth, differentiation, motility, invasion, apoptosis, and can regulate important signal transduction, immune response, DNA damage modulation, and immune cell pluripotency. It has an important biological function in the prognosis and survival of patients. Substantial evidence supports that deregulated lncRNAs are important in tumorigenesis as well as tumor progression. lncRNAs have been reported extensively in the regulation of pathophysiological functions *via* mechanisms including genetic imprinting, chromatin remodeling, histone modification, nuclear transport, transcriptional activation, transcriptional disruption, and cell cycle modulation (*Chen et al., 2015*; *Hadji et al., 2016*; *Quinn & Chang, 2016*; *Hsia et al., 2018*).

Previous investigations have affirmed that stress-stimulated long non-coding RNA (lncRNA), LINC02657, or LASTR (a lncRNA linked to splicing modulation of SART3), is upregulated in hypoxic breast cancer, a process that is critical for LASTR-positive triple-negative breast tumor. Growth is critical. Additional studies have shown that LASTRi expression is elevated in numerous forms of epithelial malignancies due to stress-induced activation of the JNK/c-JUN pathway. We also ascertained that LASTR enhances splicing efficiency by regulating the link between SART3 and U4 and U6 small nuclear ribonucleoproteins(snRNPs) in spliceosome cycling. Concurrently, at the microenvironment level, the role of lncRNAs as tumor intracellular factors are involved in mediating and controlling a variety of interactions between the immune system and malignant cells, and important mechanisms of immune responses to cancer. Many tumor-associated lncRNAs are considered to be tumor cell-intrinsic factors. They can also be tumor-cell-extrinsic factors, where both regulate cancerous cells to evade immune surveillance (*Jao et al., 2018*). Tumor-associated lncRNAs are involved in cancer immunoregulatory oncogenes or tumor inhibitory genes and have an essential function in immunotherapy resistance. However, the mechanism *via* which LncRNAs function in LUAD is yet elucidated, and it is proposed to be utilized as novel biomarkers to anticipate the prognosis of patients. Furthermore, lncRNAs can be utilized not only as a successful biomarker but also as a therapeutic target for diagnosis or prognosis due to high tissue specificity, high sefficiency, and increased stability.

In fact, there are few lncRNAs for thyroid tumors that have been discovered up to the present time. There is no research on the link between the expression and biological function of LASTR with the incidence and prognosis of lung cancer. The aim of our article was to predict the biological processes of LASTR followed by several software tools and bioinformatic analyses. We used the TCGA-LUAD level 3 RNA sequencing database to carry out genome-wide analyses and differentially prognostic caused by molecular features of LASTR. We found the expression of LASTR to be remarkably upregulated in LUAD tissues in contrast with normal tissues. We constructed tightly multiple biomolecular

interaction networks according to molecular characteristics of LASTR. In particular, the potential mediated molecular mechanisms of crosstalk among different infiltrating immune cell subsets were explored. Overall, our findings suggest that LASTR represents an active promising prognostic biomarker that regulates the immune microenvironment in LUAD.

## MATERIALS AND METHODS

### Data pre-processing and sample selection

The published data incorporated in this investigation came from The Cancer Genome Atlas (TCGA; https://portal.gdc.cancer.gov/repository). We download the level 3 RNAseq gene expression profile of LUAD patients in TPM format of TCGA as well as GTEx. In total 535 TCGA-LUAD samples and 59 normal samples were included as controls. Meanwhile, clinical data of LUAD, on the other hand, were from the TCGA database. The clinical variables included in the study were age, gender, TNM stage, residual tumor, pathologic stage, primary therapy outcome, disease-specific survival (DSS), overall survival (OS), and progression-free survival (PFS). subsequent prognostic analyses were performed based on subgroups of these variables. Patients with complete transcriptome data and with complete demographics were selected for follow-up analysis.

### Patient characteristics evaluation

We acquired LASTR expression samples from TCGA with an intention of determining its diagnostic value. On the other hand, the receiver operating characteristic (ROC) curve was created to estimate biomarkers that anticipate the prognostic survival of LUAD patients. Utilizing multivariate logistic regression, we generated a nomogram utilizing the 'rms' package in R (https://cran.r-project.org/web/packages/rms/index.html). Furthermore, we ascertained the concordance index (C-index) and contrasted the nomogram-predicted estimates with the aid of Kaplan–Meier survival probability estimates.

### LASTR Expression Level and differential gene expression analysis

With the median value aforementioned above, LUAD individuals in TCGA dataset were classified into high- and low- LASTR subgroups. To ascertain LASTR expression between tumor ($n = 535$) and non-tumor lung tissue (N = 59) samples, R language was used. The 'limma' package identified differential genes (DEGs) in the two LASTR subgroups. The False discovery rate (FDR) approach corrected the findings at adj. $P < 0.05$ level. The |log$_2$ fold change(FC)|>2 and (FDR) <0.05 criteria were utilized for screening. Additionally, Gene Ontology (GO), as well as Kyoto encyclopedia of genes and genomes (KEGG) enrichment analyses, were done with help of the 'clusterProfiler' package in R.

### Stemness-based classification determination

Consensus clustering, which is a non-monitored class discovery approach, was utilized to establish a new stemness-based classification through the 'ConsensusClusterPlus' R package. At least 1,000 repetitions were done during this clustering process by subsampling 80% of items, stratifying every subsample into multiple groups employing the k-means algorithm. The consensus matrix (CM) plot, as well as the cumulative distribution function

(CDF) plot, ascertained the optimal number of the clusters. Afterward, Kaplan–Meier (K–M) curve was performed to appraise the OS of various stemness subtypes.

### Unsupervised clustering of LASTR

We extracted a sample of 360 lncrnas with strong expression (mean FPKM $\geq$1) and highly variable ($\geq$95 FPKM percentiles of variance) of the LASTR gene from the data matrix in TCGA-LUAD. We identified sample groups with similar abundance profiles by unsupervised concordance clustering using consusclusterplus (CCP) v1.24.0. Calculations were performed using Pearson correlation, partitioning around an intermediate (Pam), 10,000 iterations, and a random 95% of the gene fraction in each iteration. We chose a five-cluster solution. To generate a rich heat map, we identified lncrnas with a mean FPKM $\geq$5 and a Sam multiple class Q value $\leq$0.01 in unsupervised clusters (see differential abundance below) , transformed by Log10(FPKM +1) for each row of the matrix, the pheatmap R package (v1.0.2) was then used to scale and cluster rows only using Pearson correlation distance metric and Ward clustering.

## Correlation of LASTR expression with survival prognosis and clinical features

The Cox proportional hazards model was utilized for carrying out a univariate analysis of disease prognosis. To ascertain independent prognosis anticipation factors, all substantial variables on the univariate Cox regression analysis ($P < 0.1$) were undergone multivariate Cox regression analysis. We thereafter did Cox regression analysis to ascertain the predictive power of various clinical variables and LASTR expression on the prognosis of LUAD to point out independent prognostic factors for TCGA-LUAD. Cox proportional hazard models as well as Kaplan–Meier plotter analysis were employed to ascertain t he link between LASTR expression and survival outcomes (OS, Progress Free Interval (PFI), and Disease Specific Survival (DSS)).

## Tumor immune infiltrating feature identification among LUAD patients

An analytical tool, CIBERSORT, was employed to import unnormalized RNA-Seq data of LUAD individuals and avail an investigation of the relative abundance of 22 immune-related cell types that were in a mixed cell population. The CIBERSORT (Cell type Identification By Estimating Relative Subsets of RNA Transcripts) was utilized to compute the relative infiltration rates of 22 tumor-infiltrating immune cells (TIICs) in TCGA samples and to determine the tumor-infiltrating immune cells (TIICs) between various groups. Samples with $p > 0.05$ were eliminated. Furthermore, the chosen samples were categorized into two groups as per the median expression value of LASTR, and the difference in lymphocytes between the two groups was analyzed.

## Creation and validation of the stemness-based classifier using several machine learning methods

A total of 535 LUAD patients were enrolled in the investigation and they had complete clinical information. The classifier was standardized within a range of 0 to 1. The prediction

capacity on the basis of the multivariate logistic model was ascertained by the time-dependent ROC curve, which additionally evaluated the classifier's optimal threshold value, with the evaluation of the AUC using the 'pROC' package.

## Co-expression genes of LASTR

We applied the Spearman correlation coefficient (PCC) to determine the degree of co-expression between the two genes. The differential genes between the two groups were characterized by the Spearman correlation coefficient with the correlation coefficient $r > 0.5$ as the threshold.

## Gene Set Enrichment Analysis

Gene Set Enrichment Analysis (GSEA) is a software that utilizes a computation approach to determine if a preset collection of genes reveals statistical significance between two phenotypes. The GSEA analysis investigated the differential pathways as well as biology functions between the two LASTR groups (version 4.0.3) in the TCGA-LUAD cohort. The reference gene set was c2.cp.KEGG.v7.1.symbols.gmt and $p < 0.05$ indicated a statistically significant difference.

## Quantitative reverse transcription-polymerase chain reaction (qRT-PCR)

TRIzol reagent (Sigma-Aldrich, St. Louis, MO, USA) and FastStart Universal SYBR®Green Master (Roche, Indianapolis, IN, USA) was used to extracted total RNA. The sequences of target gene primer pairs involved in the study are as follows:

| Gene | Forward primer sequence (5-3) | Reverse primer sequence (5-3) |
|------|-------------------------------|-------------------------------|
| LASTR | AGTGGGTGAAGTCCTGGTT | GGCTGAAGGGTTTAGATG |
| GAPDH | AATGGGCAGCCGTTAGGAAA | GCCCAATACGACCAAATCAGAG |

## Cell culture and transient transfection

The A549 and H1299 lung cancer cell lines were provided by the cell culture bank under the Chinese Academy of Sciences, which were then incubated in F12 and DMEM supplemented with 10% fetal bovine serum (FBS) acquired from Gibco (Carlsbad, CA, USA), respectively. The cell lines were cultured at 37°C and 5% carbon dioxide. The negative control (NC) and LASTR shRNA (Obio Technology, Shanghai, China) were transfected into cells acquired from Invitrogen (Carlsbad, CA, USA). The target sequences for LASTR shRNAs were 5′-AGGGTTAATGACTCAATTTTT-3′ (LASTR sh 1), 5′-GGAAATTCAGATCATCTAAAC-3′ (LASTR sh 2) and 5′-TGC TAGTAATGACAATCATGT-3′ (LASTR sh 3). At the same time, cell culture dishes/plates, and centrifuge tubes were obtained from NEST Biotechnology Co. Ltd. (Wuxi, China)

## Transwell assay

Transwell assays for lung cancer cell (A549, H1299) migration and invasion were performed. Briefly, cells ($5 \times 10^4$) were inoculated into chambers coated (for invasion) or uncoated

with Matrigel (for migration). The base medium used is composed of DMEM and the top layer is SFM agar (*Lu et al., 2022*). After a incubation period of 24 hours, it was stained with 0.1% crystal violet cells and fixed with 4% paraformaldehyde. Finally, the cell count under light microscope was completed.

### CCK8 assay
The density of inoculated cells in 96-well plates was 1,000 cells per well. CCK8 reagent (Beyotime, Shanghai, China) was then added to each well, and after 1.5 hours of cell culture, the absorbance at 450 nm was measured.

### Statistical analysis
The Spearman correlation test was performed to investigate the link between two variables. Whereas the Chi-square test was carried out to contrast the various subgroup's categorical and pairwise features. In order to compare the various subgroups' ordinal as well as non-normally distributed data, a Wilcoxon test was adopted. Kruskal-Wallis was used to test the dependent or continuous variables of two or more ordered classes (*Hadji et al., 2016*). The R program was used (Version 4.1.0). The two-tailed test was used for statistical tests, and $p \leq 0.05$ was considered statistically significant.

## RESULT

### High LASTR expression in LUAD
Firstly, we ascertained the distribution of LASTR in various parts of the human body (Fig. 1A). In order to ascertain the status of LASTR expression in LUAD individuals, we compared the expression levels of that in normal lung tissues. The findings affirmed that the LASTR expression level was substantially elevated in LUAD tissues and corresponding normal lung tissues ($p < 0.001$) (Figs. 1B–1C). Further, to value LASTR's diagnostic effectiveness in LUAD, we developed ROC curves of the area under the ROC curve (AUC). ROC analyses affirmed that LASTR expression can be one substantial parameter to distinguish between normal and malignant tissues with an area under the ROC curve (AUC) of 0.922 (95% CI [0.900–0.944]) (Fig. 1D).

### Unsupervised clustering using LASTR identifies two prognostic patient subgroups
We additionally conducted a consensus clustering analysis to generate a robust genomic subtype of LUAD individuals linked to LASTR expression levels. A Sankey diagram was used to show whether the consistent clustering grouping related to LASTR expression could effectively identify LUAD patients. Unsupervised clustering of all samples of the TCGA-LUAD dataset based on LASTR expression. To ascertain the optimal number of gender-specific clusters, we changed the number of clusters from two to four each and analyzed the cumulative density function (CDF) curves of the consensus matrix (Figs. 2A–2C). The optimal number of clusters based on the AUC values of the consensus distribution function (CDF) plot was discovered to be 2 (Figs. 2D–2E). We matched the identified two subgroups (cluster A: $n= 375$, cluster B: $n= 122$) with normal and tumor tissues of the TCGA-LUAD dataset by the Sankey diagram, and found that normal tissues

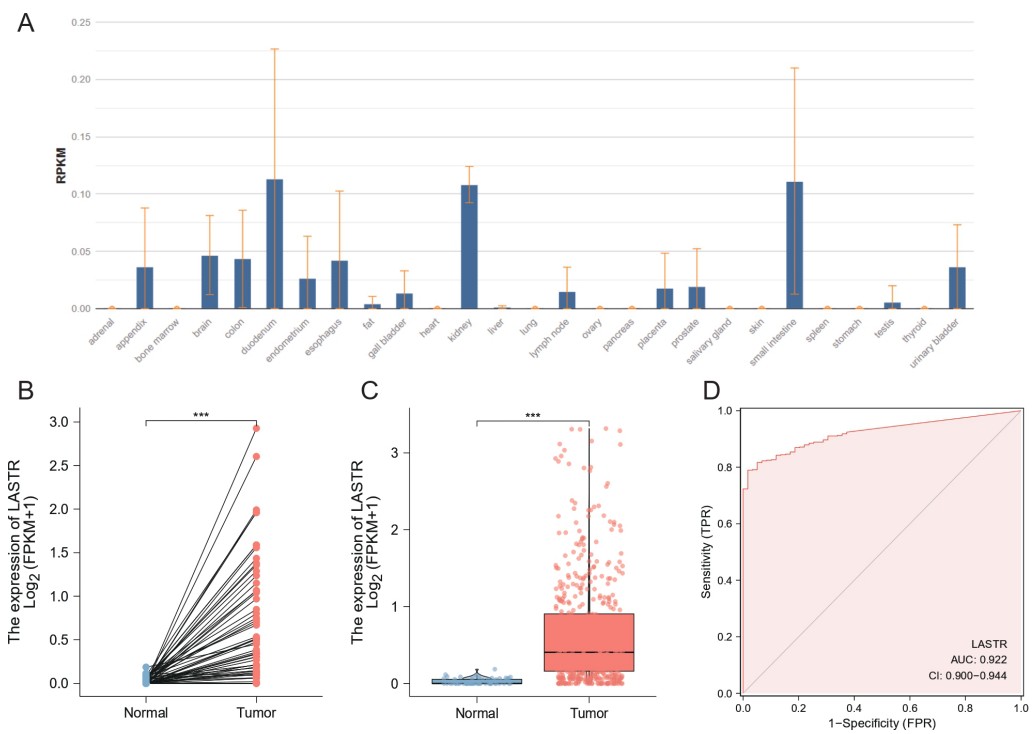

**Figure 1** **LASTR expression level and diagnostic efficacy for LUAD.** (A) LASTR expression levels in different types of tissues. (B) Paired differential expression analysis between LUAD samples and paired normal controls. (C) TCGA-LUAD database verified that LASTR expression was significantly upregulated in LUAD ($n = 497$) compared with normal kidney tissues ($n = 52$). (D) ROC curves showed high AUC values in TCGA-LUAD, showing the ability of LASTR expression to discriminate between LUAD and normal tissue samples. $^*p < 0.05$, $^{**}p < 0.01$, $^{***}p < 0.001$.

can be completely differentiated (Fig. 2F). It can be considered that the LASTR-based unsupervised clustering grouping can well discriminate LUAD.

## High LASTR expression is linked to the poor OS in patients with LUAD

We performed logistic analysis and cox regression, independent prognostic, survival, and clinical correlation analyses. Logistic regression identified high-LASTR expression independent predictors for early death. Pearson correlation analysis analyzed the link between LASTR and related mRNAs. The Kaplan–Meier method revealed the prognostic significance of the lncRNAs, whereas the log-rank test analyzed survival time.

Then, we counted and analyzed the grouping information of 535 samples obtained from the TCGA-LUAD dataset, including age, gender, pathologic stage, TNM stage, primary therapy outcome, OS, PFI event, DSS, and residual tumor. The demographic and clinical characteristics of the individuals in TCGA were summarized in Table 1.

We performed OS analysis of LASTR expression in relation to various subgroups of LUAD patients (Fig. 3A). Furthermore, we performed a survival analysis on LASTR utilizing the "survival" and "survminer" packages in R. Survival analysis ascertained that high LASTR expression was remarkably associated with worse outcomes in LUAD

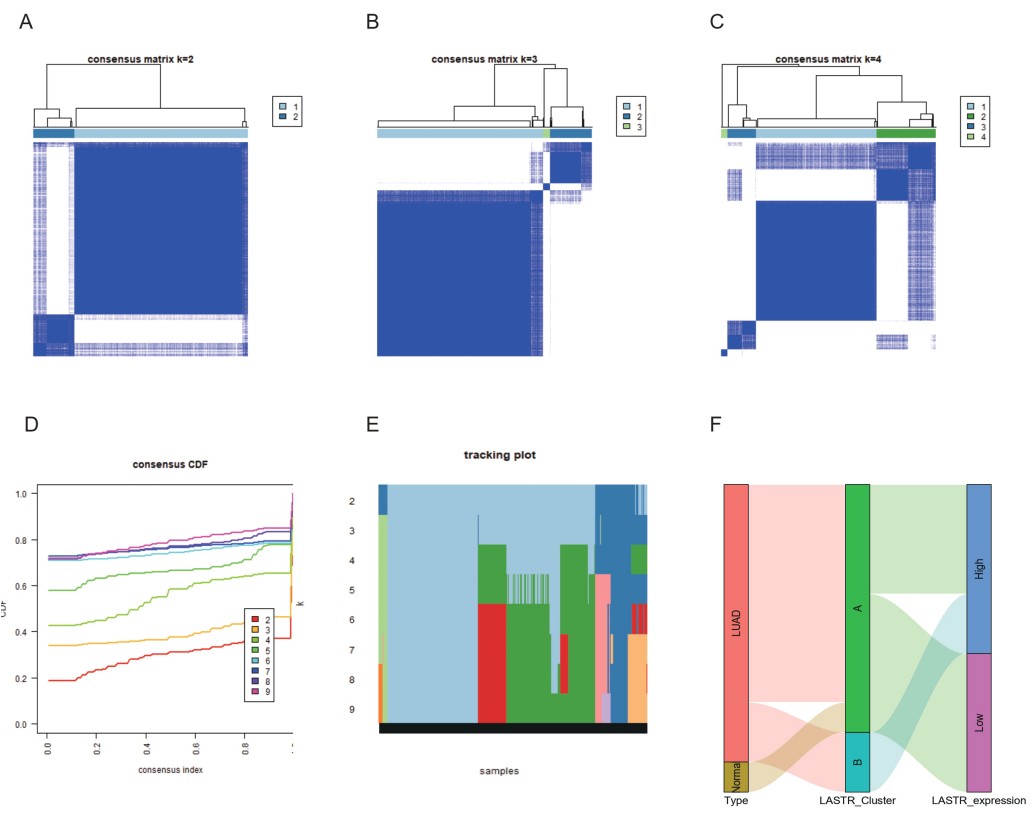

**Figure 2** **A ConsensusCluster evaluation based on LASTR expression disclosed the existence of two primary clusters of all samples.** (A–C) Specific clusters from 2 to 4 of each and examined the CDF curves of the consensus matrix; (D) CDF plot to find the smallest k; (E) Sample cluster distribution map; (F) Sankey diagram matched the 2 cluster subgroups with normal and tumor tissues of the TCGA-LUAD.

individuals. Survival analysis affirmed that high LASTR expression was remarkably linked to poor OS ($P = 0.001$), progression-free interval (PFI) ($P = 0.024$), and DSS ($P = 0.024$) (Figs. 3B–3D). High LASTR expression in LUAD tissue samples was substantially linked to worse OS, HR = 1.66 (1.24–2.22), $P = 0.001$ (Fig. 3B), progression-free interval HR = 1.36 (1.04–1.77), $P = 0.024$ (Fig. 3C), DSS HR = 1.53 (1.06–2.20), $P = 0.024$ (Fig. 3D).

In addition, we performed an exploratory analysis of overall survival in clinical subgroups. In subgroup analyses of prespecified clinical variables, in further survival analyses, an OS disadvantage in the high LASTR expression group was observed in most clinically relevant subgroups quantitative analysis of Kaplan–Meier overall survival curves manifested a remarkable link between high LASTR expression and poor OS in the TNM stage, pathologic stage and tumor status clinicopathological subgroups. Subgroup survival analysis utilizing clinicopathological characteristics established that samples with high LASTR expression exhibited a disadvantage in overall survival status based on the pathological characteristics of high TNM stage, high-grade pathological stage, and tumor status (Fig. 4). Subgroup analysis showed that T stage-T1 & T2 & T3 & T4, HR = 1.70 (1.26–2.29), $p < 0.001$; M stage-M0 & M1 HR = 1.53 (1.10–2.13) $p = 0.011$; N stage-N1

**Table 1  Clinical characteristics of the LUAD patients.**

| Characteristic | Type | Low expression of LASTR | High expression of LASTR | adj.P |
|---|---|---|---|---|
| n | | 267 | 268 | |
| T stage, n (%) | T1 | 106 (19.9%) | 69 (13%) | 0.003 |
| | T2 | 132 (24.8%) | 157 (29.5%) | |
| | T3 | 19 (3.6%) | 30 (5.6%) | |
| | T4 | 7 (1.3%) | 12 (2.3%) | |
| N stage, n (%) | N0 | 186 (35.8%) | 162 (31.2%) | 0.010 |
| | N1 | 39 (7.5%) | 56 (10.8%) | |
| | N2 | 28 (5.4%) | 46 (8.9%) | |
| | N3 | 0 (0%) | 2 (0.4%) | |
| M stage, n (%) | M0 | 185 (47.9%) | 176 (45.6%) | 0.205 |
| | M1 | 9 (2.3%) | 16 (4.1%) | |
| Pathologic stage, n (%) | Stage I | 167 (31.7%) | 127 (24.1%) | 0.005 |
| | Stage II | 53 (10.1%) | 70 (13.3%) | |
| | Stage III | 33 (6.3%) | 51 (9.7%) | |
| | Stage IV | 10 (1.9%) | 16 (3%) | |
| Primary therapy outcome, n (%) | PD | 33 (7.4%) | 38 (8.5%) | 0.818 |
| | SD | 18 (4%) | 19 (4.3%) | |
| | PR | 4 (0.9%) | 2 (0.4%) | |
| | CR | 166 (37.2%) | 166 (37.2%) | |
| Gender, n (%) | Female | 149 (27.9%) | 137 (25.6%) | 0.317 |
| | Male | 118 (22.1%) | 131 (24.5%) | |
| OS event, n (%) | Alive | 184 (34.4%) | 159 (29.7%) | 0.026 |
| | Dead | 83 (15.5%) | 109 (20.4%) | |
| DSS event, n (%) | Alive | 196 (39.3%) | 183 (36.7%) | 0.390 |
| | Dead | 56 (11.2%) | 64 (12.8%) | |
| PFI event, n (%) | Alive | 157 (29.3%) | 152 (28.4%) | 0.689 |
| | Dead | 110 (20.6%) | 116 (21.7%) | |
| Residual tumor, n (%) | R0 | 170 (45.7%) | 185 (49.7%) | 0.185 |
| | R1 | 7 (1.9%) | 6 (1.6%) | |
| | R2 | 0 (0%) | 4 (1.1%) | |
| Age, meidian (IQR) | | 67 (60, 72) | 65 (58, 72) | 0.376 |

& N2 & N3 & N0 HR =1.66 (1.23–2.24) $p < 0.001$; Pathological stage-Stage I & Stage II & Stage III & Stage IV HR = 1.69 (1.26–2.27) $p < 0.001$; Tumor status-with tumor HR = 1.47 (1.00–2.14) $p = 0.048$.

## Multivariate analysis with logistic regression

Survival analysis was done with Cox regression as well as logistic regression. They are procedures where the response variable is either a dichotomous variable or the integration of a response variable and a continuous variable. Logistic regression is a binary model that is widely utilized for object classification in addition to pattern recognition. The results of a binary logistic regression to identify independent predictors for early death can be viewed in Table 2, which can reflect the degree of relationship between each clinical variable. The

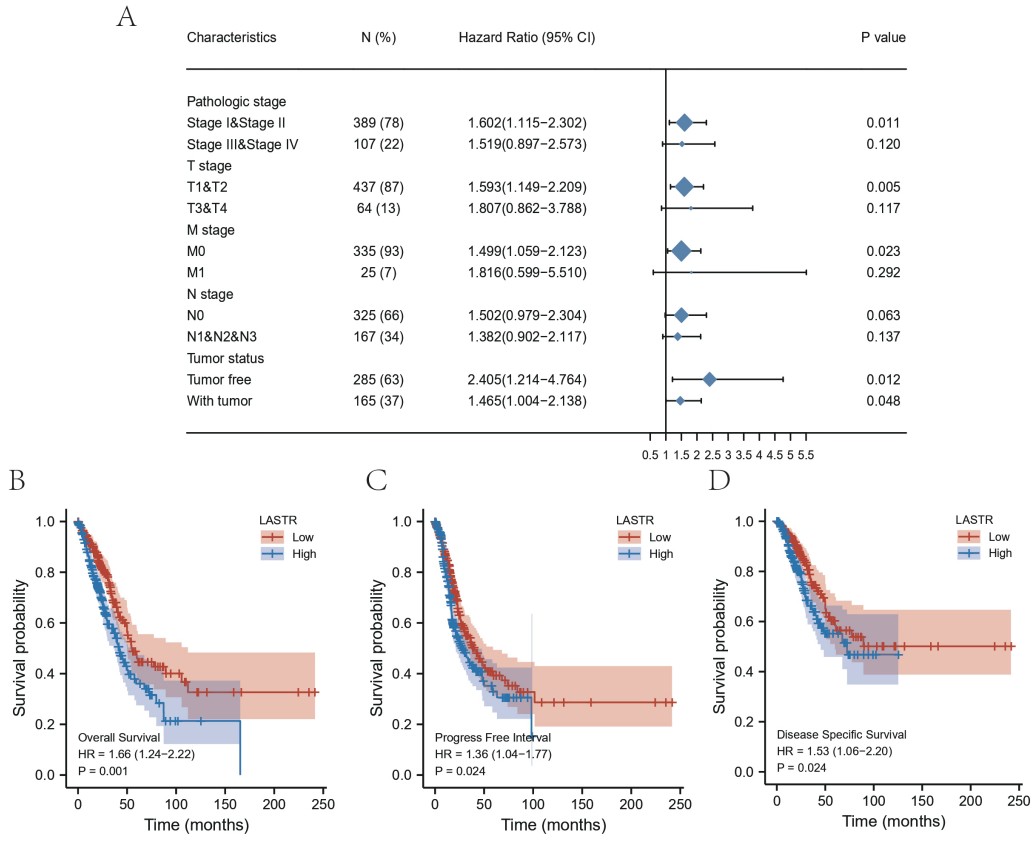

**Figure 3** **(A) A forest plot of the potential prognostic variables and Kaplan–Meier curve for survival in LUAD; Kaplan–Meier curves of (B) OS, (C) PFI and (D) DSS for high and low LASTR expression groups.** (A) A forest plot of the potential prognostic variables and Kaplan–Meier curve for survival in LUAD; Kaplan–Meier curves of (B) OS, (C) PFI and (D) DSS for high and low LASTR expression groups.

link established between LASTR expression and clinical features in individuals with LUAD suggested that a high expression level of LASTR was greatly correlated with T stage (T3 & T4 *vs.* T1 & T2) ($p = 0.046$), N stage (N1 & N2 & N3 *vs.* N0) ($p = 0.002$), Pathologic stage (Stage III & Stage IV *vs.* Stage I & Stage II) ($p = 0.011$). Nonetheless, the elevated LASTR expression level was not remarkably linked to any other clinical characteristics. In order to further explore the clinical status of LASTR affecting the prognosis of lung adenocarcinoma, we included clinical information such as TNM stage, pathological stage, main efficacy evaluation, gender, age, race, tumor site, smoking status and the expression level of LASTR in the TCGA-LUAD database into Cox analysis. It was found that LASTR could be used as an independent factor affecting the overall survival prognosis of LUAD patients, and together with the Primary therapy outcome and Tumor status could affect the overall survival outcome of LUAD patients (Table 3).

## LASTR revealed a significant link to various TIICs in LUAD

To ascertain the link between LASTR expression and TIICs, we made an estimate of the proportions of 22 varied immune cell types utilizing the CIBERSORT algorithm. Besides,

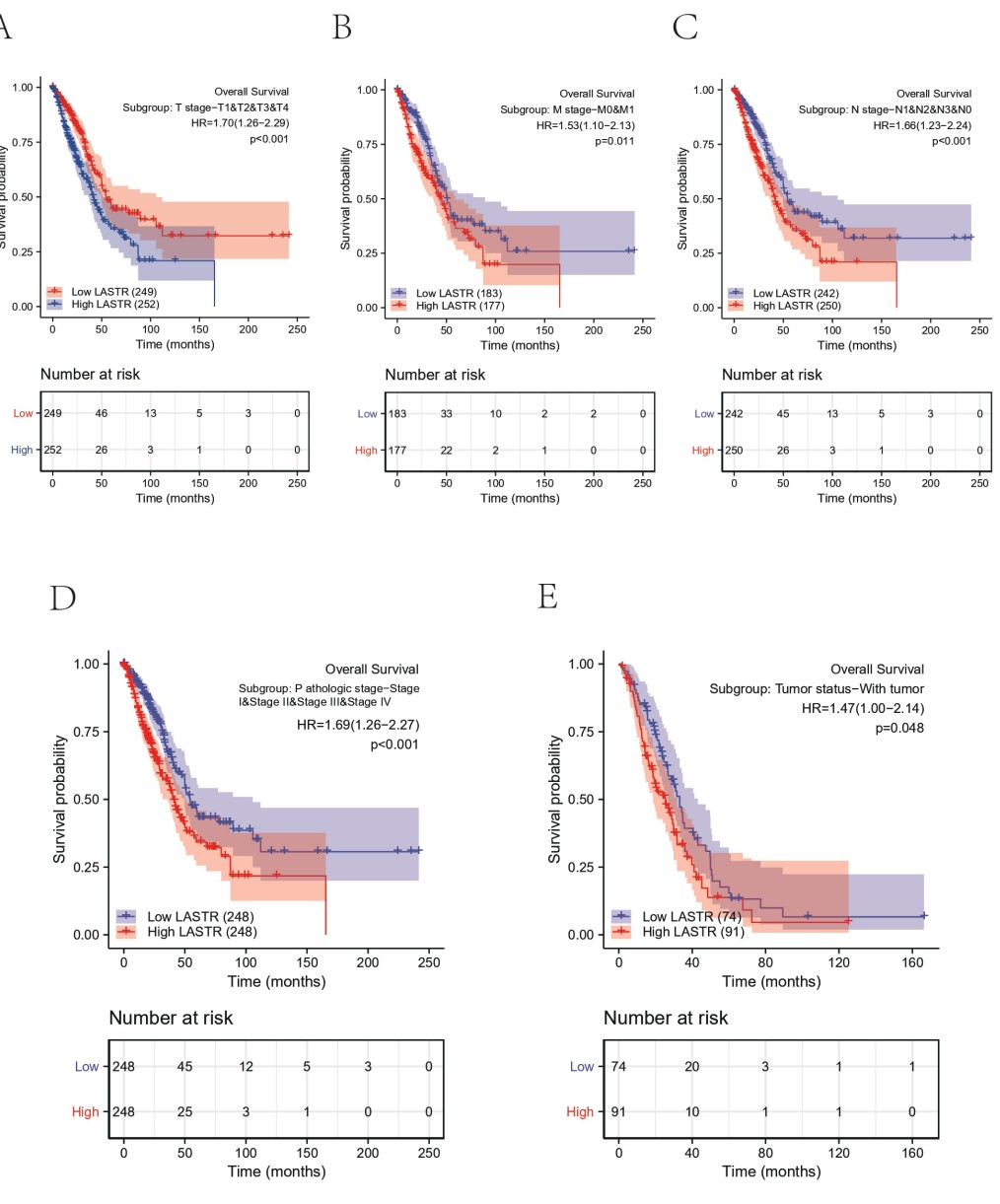

**Figure 4 OS analysis of Kaplan–Meier curves between different clinical subgroups in LUAD.** The survival curve in each of the clinicopathologic subgroups of TCGA-LUAD, including (A) T stage, (B) M stage, (C) N stage, (D) Pathologic stage, and (E) Tumor status.

CIBERSORT was utilized to conduct immune cell deconvolution. We identified immune cell types that depicted various degrees of connection with expression levels of the LASTR. We Statistically analyzed these differences, which presented a comparable link to immune cells. From the difference in immune infiltrate results, we contrasted the difference of TIICs between the two LASTR groups utilizing the median threshold value of LASTR expression. We know that the baseline characteristics of patients between LASTR high and low expression groups. The proportion of Mast cells, TFH, Tcm, T helper cells, iDC,

**Table 2  Logistic analysis of the association between LASTR expression and clinical characteristics.**

| Characteristics | Total (N) | Odds Ratio (OR) | *P* value |
|---|---|---|---|
| T stage (T3 & T4 *vs.* T1 & T2) | 532 | 1.701 (1.016–2.897) | 0.046 |
| N stage (N1 & N2 & N3 *vs.* N0) | 519 | 1.782 (1.231–2.594) | 0.002 |
| M stage (M1 *vs.* M0) | 386 | 1.869 (0.820–4.519) | 0.146 |
| Pathologic stage (Stage III & Stage IV *vs.* Stage I & Stage II) | 527 | 1.740 (1.138–2.685) | 0.011 |
| Age (>65 *vs.* <=65) | 516 | 0.756 (0.534–1.068) | 0.113 |

**Table 3  Univariate and multivariate Cox regression analyses of clinical characteristics associated with overall survival.**

| Characteristics | Total | Univariate analysis | | Multivariate analysis | |
|---|---|---|---|---|---|
| | | HR (95% CI) | *P* value | HR (95% CI) | *P* value s |
| T stage (T2 & T3 & T4 *vs.* T1) | 501 | 1.668 [1.184–2.349] | 0.003 | 1.311 [0.765–2.246] | 0.324 |
| N stage (N1 & N2 & N3 *vs.* N0) | 492 | 2.606 [1.939–3.503] | <0.001 | 1.572 [0.778–3.176] | 0.208 |
| M stage (M1 *vs.* M0) | 360 | 2.111 [1.232–3.616] | 0.007 | 1.003 [0.433–2.326] | 0.994 |
| Pathologic stage (Stage II & Stage III & Stage IV *vs.* Stage I) | 496 | 2.975 [2.188–4.045] | <0.001 | 0.908 [0.414–1.993] | 0.810 |
| Primary therapy outcome (PD & SD & PR *vs.* CR) | 419 | 2.818 [2.004–3.963] | <0.001 | 1.962 [1.244–3.096] | 0.004 |
| Gender (Male *vs.* Female) | 504 | 1.060 [0.792–1.418] | 0.694 | | |
| Age (>65 *vs.* <=65) | 494 | 1.228 [0.915–1.649] | 0.171 | | |
| Race (White *vs.* Asian & Black or African American) | 446 | 1.422 [0.869–2.327] | 0.162 | | |
| Anatomic neoplasm subdivision (Right *vs.* Left) | 490 | 1.024 [0.758–1.383] | 0.878 | | |
| Anatomic neoplasm subdivision2 (Peripheral Lung *vs.* Central Lung) | 182 | 0.913 [0.570–1.463] | 0.706 | | |
| number pack years smoked (>=40 *vs.* <40) | 345 | 1.038 [0.723–1.490] | 0.840 | | |
| Smoker (Yes *vs.* No) | 490 | 0.887 [0.587–1.339] | 0.568 | | |
| Tumor status (With tumor *vs.* Tumor free) | 450 | 6.211 [4.258–9.059] | <0.001 | 6.288 [3.709–10.661] | <0.001 |
| LASTR (High *vs.* Low) | 504 | 1.666 [1.241–2.237] | <0.001 | 1.740 [1.109–2.732] | 0.016 |

eosinophils, DC, and B cells were significantly reduced in the high-LASTR group in the TCGA-LUAD cohort while the infiltration level of NK CD56dim cells, Th2 cells, Tgd were significant increased (Figs. 5A–5B).

In order to further assess the relationship of candidate gene expression with immune-oncological mechanisms, we employed correlation analysis between LASTR expression and immune infiltration level for LUAD. There were significantly positive correlated with abundance of NK CD56dim cells (Spearman $r = 0.162$, $P < 0.001$), Th2 cells (Spearman $r = 0.283$, $P < 0.001$), Tgd (Spearman $r = 0.139$, $P = 0.001$). Negative values of r show a negative correlation between data sets. There were also negatively correlated between LASTR expression and Tcm (Spearman $r = -0.163$, $p < 0.001$), Mast cells (Spearman $r = -0.241$ $P < 0.001$), Eosinophils (Spearman $r = -0.159$, $p < 0.001$) (Figs. 5C–5H). These findings ascertained that LASTR exhibited an obvious connection with several infiltrating lymphocytes and further research on it is worthy.

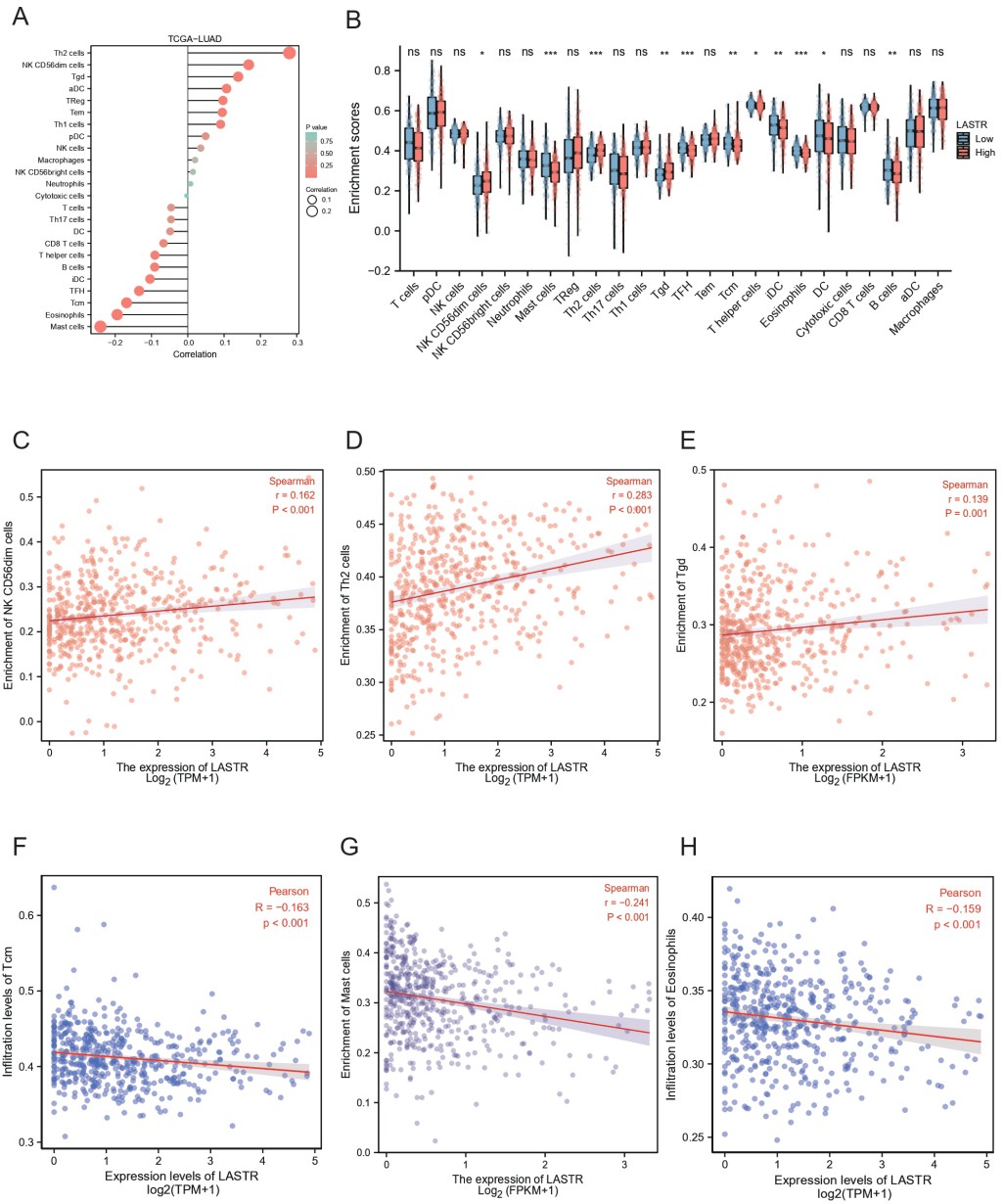

**Figure 5  Analysis of tumor-infiltrating immune cells in high as well as low expression of LASTR.** (A) The lollipop chart shows the statistical correlation of immune cell infiltration between the two expression groups of LASTR; (B) Box plots represent the differences in the infiltration levels of 22 different immune cell types estimated by the CIBERSORT algorithm between two LASTR expression groups; (C–E) LASTR was positively correlated with NK CD56dim cells (Spearman $r = 0.162$, $P < 0.001$), Th2 cells (Spearman $r = 0.283$, $P < 0.001$), Tgd (Spearman $r = 0.139$, $P = 0.001$), respectively; (F–H) LASTR was positively correlated with Tcm (Pearson $R = -0.163$, $p < 0.001$), Mast cells (Spearman $r = -0.241$ $P < 0.001$), Eosinophils (Pearson $R = -0.159$, $p < 0.001$) were negatively correlated.

## Co-expressed genes with LASTR

To obtain co-expression genes closely associated with LASTR expression, the median expression level of LASTR was utilized to cluster TCGA-LUAD samples into high as

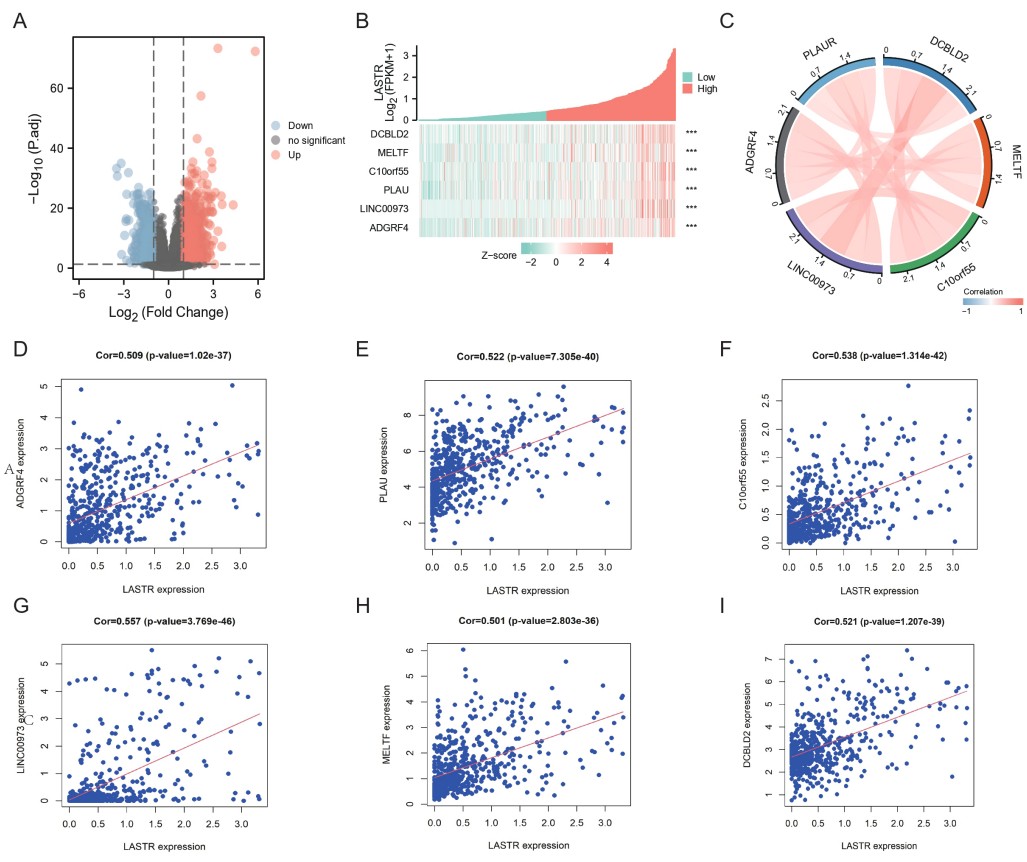

**Figure 6** **Correlation evaluation of co-expressed genes closely linked to LASTR.** (A) Volcano plot demonstrating DEGs based on LASTR expression of LUAD from TCGA. The red dots denotes up-regulated genes, whereas the blue dots stands for down-regulated genes; (B) Expression profiles of Top6 co-expression genes of LASTR across TCGA LUAD samples; (C) Network diagram showing gene-gene dependencies closely related to LASTR; (D–I) Scatter plot characterization between LASTR and DCBLD2 (cor = 0.521), MELTF (cor = 0.501), C10orf55 (cor = 0.538), PLAU (cor = 0.522), LINC00973 (cor = 0.557), ADGRF4 (cor = 0.509) significantly correlated linear fits and correlations between genes.

well as low expression groups. The between-group expression differences of all genes were processed in R using the Linear Models for Microarray Analysis (Limma) package (adjusted $P < 0.05$ and absolute Log|2 FC|>2). We compared the differential genes between two expression groups of LASTR, and the differential expression results were displayed by a volcano plot (Fig. 6A). Spearman's rank tests explored the correlation between the LASTR and co-expressed genes. We analyzed their expression differences and correlations, showing them as heatmap and network graphs, and fitted the degree of linear correlation for each co-expressed gene (Figs. 6B–6C). The top six co-expressed genes with correlation coefficients greater than 0.5 were selected for additional analysis. The scatter plot exhibited the correlation coefficients between LASTR and DCBLD2 (cor = 0.521), MELTF (cor = 0.501), C10orf55 (cor = 0.538), PLAU (cor = 0.522), LINC00973 (cor = 0.557), ADGRF4 (cor = 0.509) (Figs. 6D–6I).

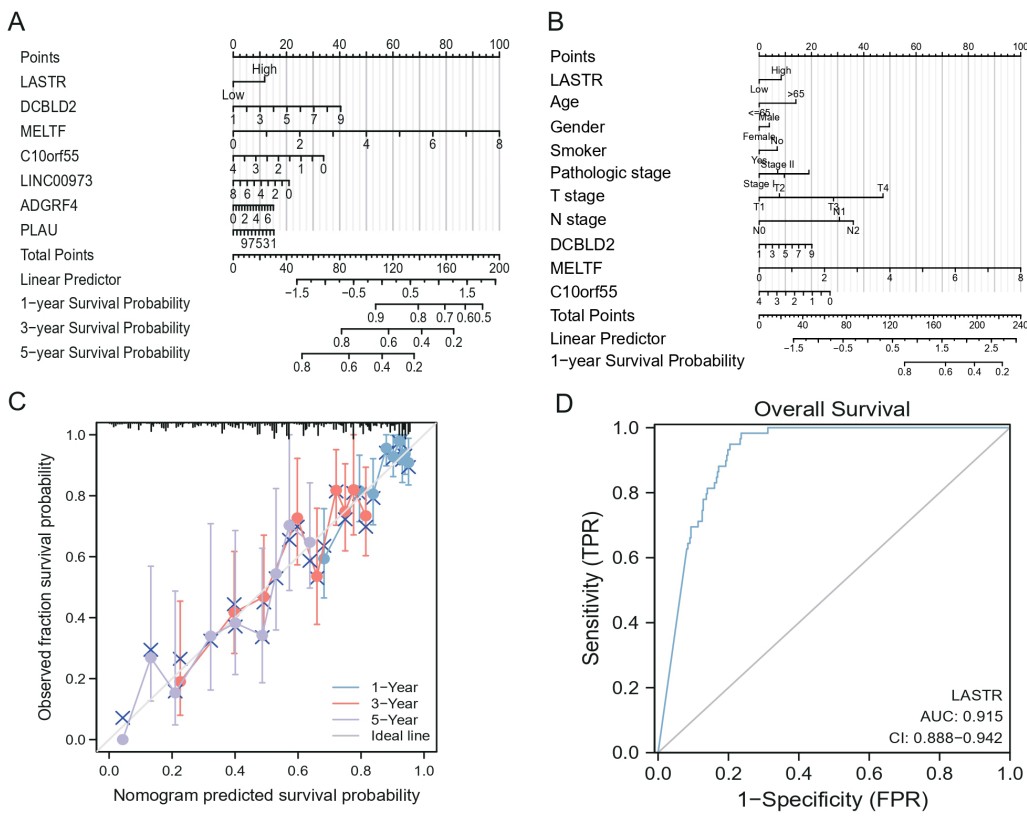

**Figure 7 Correlation between LASTR gene expressions and clinicopathological parameters in LUAD.**
(A) A nomogram of gene expression predicting 1, 3, and 5 years of patient outcomes based on LASTR expression; (B) A nomogram of clinicopathological features predicting 1, 3, and 5 years of patient outcomes based on LASTR expression; (C) The nomogram calibration plot depicts that the nomogram was calibrated well, with mean predicted probabilities for every subgroup near observed probabilities; (D) The discriminative power of high expression LASTR in predicting poor OS was analyzed by ROC curve.

## Establishment of nomogram

We developed a nomogram to anticipate OS in the TCGA-LUAD cohort by integrating the above clinical parameters. In support of the clinical use of our findings, we have constructed a nomogram for anticipating 1-, 3-, and 5-year-related gene expression and clinicopathological features affecting patient OS based on LASTR expression (Figs. 7A–7B). In addition, a calibration curve was evaluated for the fit between the established nomogram survival states and the actual survival states by the bootstrap method (1,000 replicates) (Fig. 7C). We used ROC curves to assess the discriminative power of high-expression of LASTR in predicting samples with poor OS (AUC =0.915) (Fig. 7D).

## Co-expression analysis of LASTR and enrichment analysis

We carried out GO enrichment analysis for the genes and discovered that the GO was mostly enriched in terms like GO:0070268-cornification (Fig. 8 & Table 4). In KOBAS, a full-screen view of each PathBank pathway diagram is also available, green represents down-regulated LASTR co-expressed genes, and red represents up-regulated LASTR co-expressed genes.

**Table 4   GO enrichment analysis of LASTR differentially co-expressed genes.**

| Ontology | ID | Description | p.adjust | qvalue |
|---|---|---|---|---|
| BP | GO:0070268 | Cornification | 5.55e−10 | 5.18e−10 |
| BP | GO:0031424 | Keratinization | 2.25e−07 | 2.10e−07 |
| CC | GO:0005882 | Intermediate filament | 1.40e−05 | 1.20e−05 |
| CC | GO:0045095 | Keratin filament | 1.40e−05 | 1.20e−05 |
| MF | GO:0005200 | Structural constituent of cytoskeleton | 0.002 | 0.002 |
| MF | GO:0004252 | Serine-type endopeptidase activity | 0.010 | 0.008 |

**Table 5   KEGG enrichment analysis of LASTR differentially co-expressed genes.**

| Ontology | ID | Description | p.adjust | qvalue |
|---|---|---|---|---|
| KEGG | hsa04151 | PI3K-Akt signaling pathway | 0.028 | 0.025 |
| KEGG | hsa04080 | Neuroactive ligand–receptor interaction | 0.008 | 0.007 |
| KEGG | hsa05150 | Staphylococcus aureus infection | 0.028 | 0.025 |
| KEGG | hsa04915 | Estrogen signaling pathway | 0.058 | 0.053 |

The interactive pathway viewer reveals selected pathway diagrams and emphasizes the active genes, which can identify genes that are differentially expressed in the pathways (Fig. 9 & Table 5). We showed the specific enrichment of hsa05202 transcriptional misregulation in cancer, hsa05210: colorectal cancer, hsa05226: gastric cancer, hsa04151: PI3K-Akt signaling pathway, hsa04080: neuroactive ligand-receptor interaction, and hsa04915: the estrogen signaling pathway in Pathview. KEGG enrichment analysis demonstrated that in Transcriptional misregulation in cancer (hsa05202), colorectal cancer (hsa05210), gastric cancer (hsa05226) (Figs. 9A–9C), PI3K-Akt signaling pathway (Fig. 9D), neuroactive ligand-receptor interaction (Fig. 9E), estrogen signaling pathway (Fig. 9F).

## Biological function of LASTR in cancer

We utilized Gene Set Enrichment Analysis (GSEA), which is instrumental in determining the statistical significance of a priori-defined set of genes as well as the existence of concordant differences across biological states. We set the gene set permutation in this investigation to 1.000 times in each analysis. LASTR expression level was utilized as a phenotype label. The gene sets "c2.cp.v7.2.symbols.gmt (v7.2)" was retrieved from the Molecular Signatures Database to establish gene sets that their expression is enriched or depleted in high aneuploidy cancers. We found that LASTR is involved in DNA replication, cell cycle, and immune cell infiltration pathways (Figs. 10A–10B). The downregulated gene expression demonstrates enrichment of LASTR targets present in lymphocytes, CTLA4, and other immune microenvironment-related pathways, as well as fatty acid and glutathione metabolism, and neuroactive ligand-receptor interactions (Figs. 10C–10D). Meanwhile, DNA replication, cell cycle checkpoints, cell cycle, DNA replication, cytokine receptor interactions, MTOR signaling pathway, and natural killer cell-mediated cytotoxicity were the most differentially enriched pathways in LASTR low expression samples (Figs. 10E–10F).

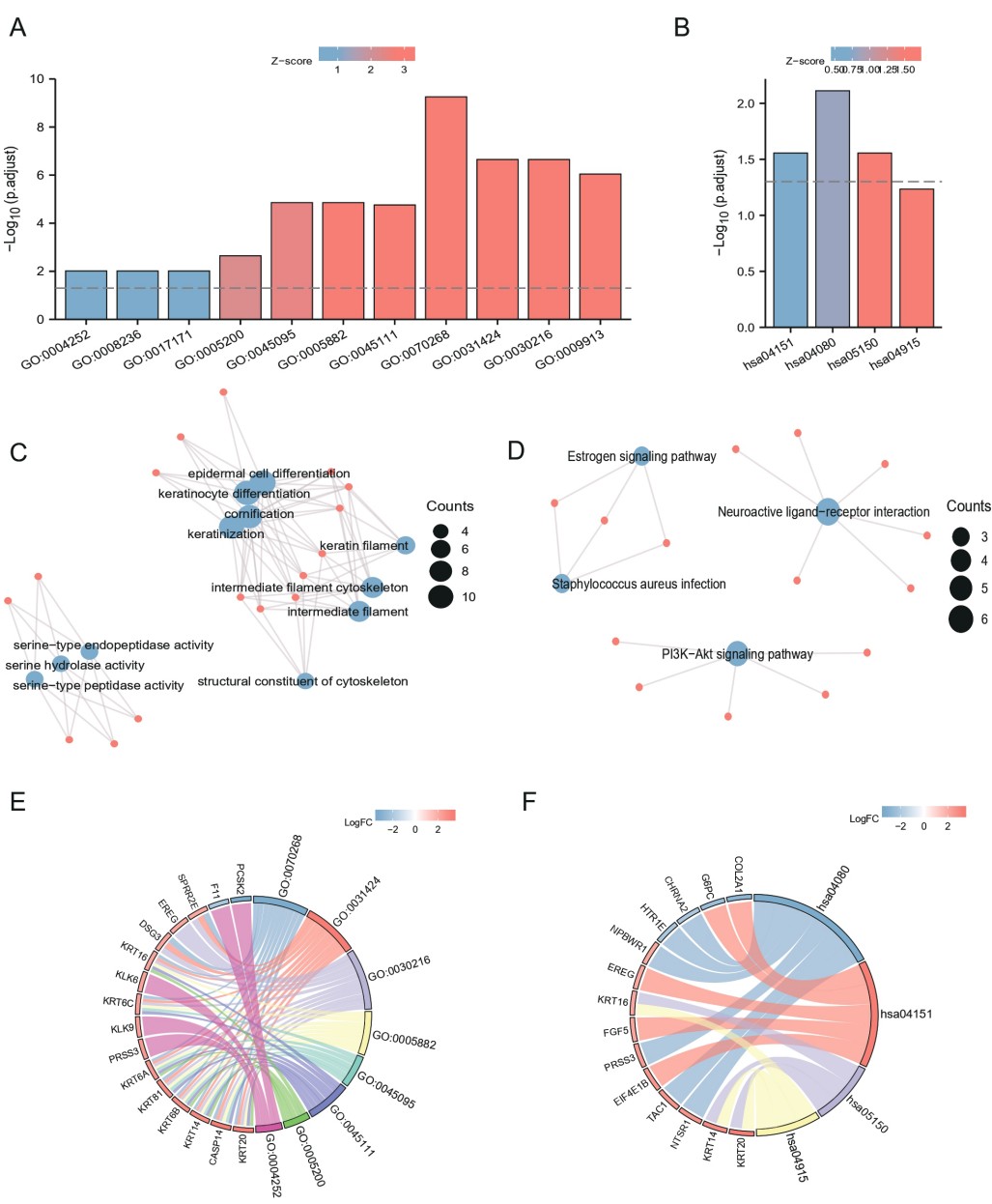

**Figure 8   GO/KEGG enrichment analysis of LASTR co-expressed genes.** (A–B) GO & KEGG enrichment histogram, (C–D) GO & KEGG enrichment network chart, (E–F)[b] GO & KEGG enrichment chord chart, the figure shows term with p.adjust <0.05. The length of the bars in the histogram represents the amount of gene enrichment, the color represents the significance, and the significance increases gradually from blue to red.

## LASTR was highly expressed in tumor tissues *in vitro*

We verified the difference of LASTR expression in tumor tissues in lung cancer A549 cell line and H1299 cell line. First, RT-PCR was used to verify the increase of LASTR in tumor tissue, which was consistent with the results obtained from bioinformatics analysis (Fig. 11A). Subsequently, detect the transfection efficiency of sh LASTR in A549 cell line

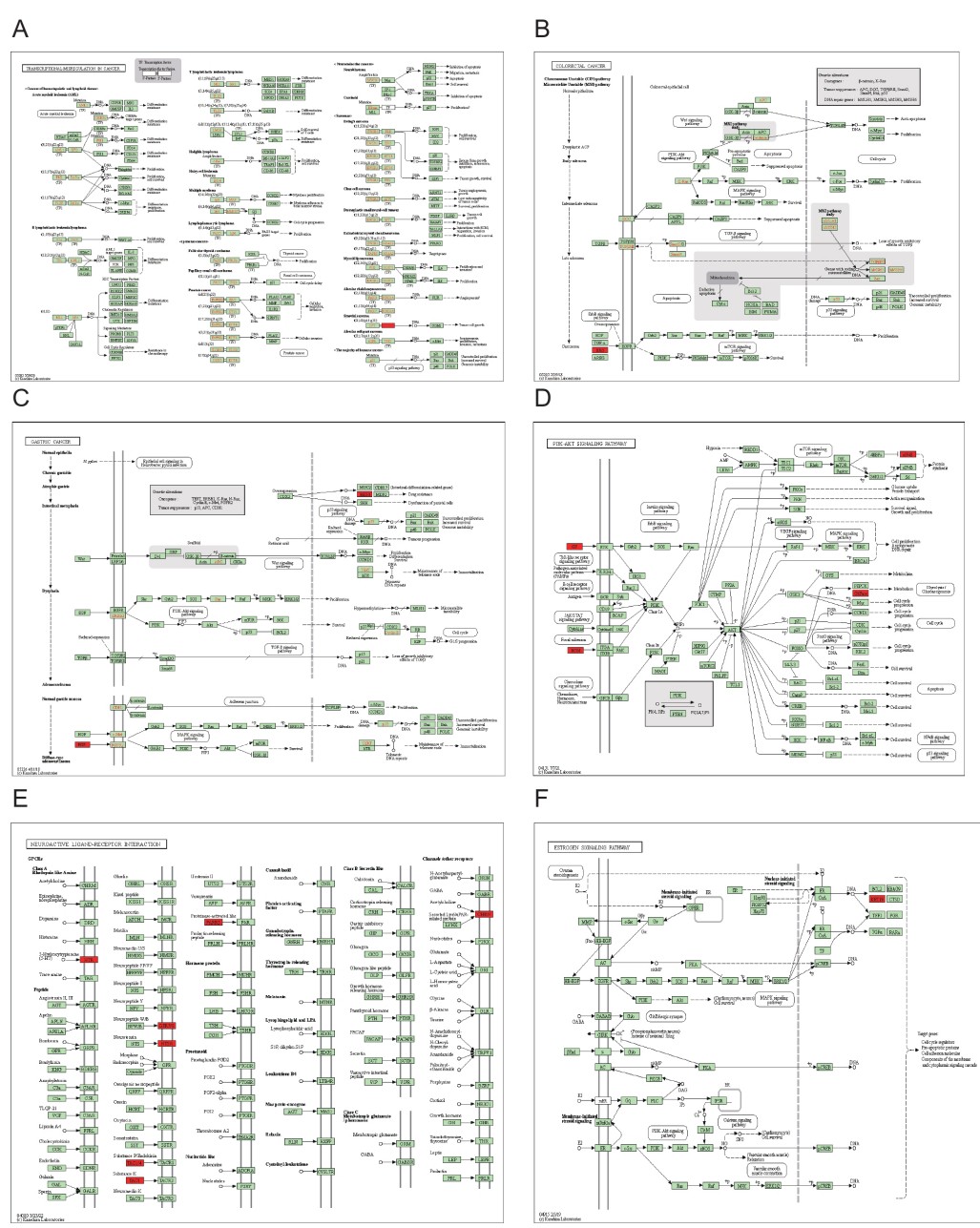

**Figure 9 Pathview diagram with differentially expressed genes.** Green represents down-regulated LASTR co-expressed genes, and red represents up-regulated LASTR co-expressed genes. (A) hsa05202, (B) hsa05210, (C) hsa05226, (D) hsa04151, (E) hsa04080, (F) hsa04915.

and H1299 cell line, and standardize the results (Figs. 11B–11D). Next, CCK8 and Transwell experiments were used to verify the malignant phenotype of tumor migration and invasion *in vitro*, and passed. Not surprisingly, the qualitative and quantitative analysis results of apoptosis flow cytometry visualization showed that the migration and invasion ability of the negative control group and LASTR knockout group decreased (Figs. 11E–11L).

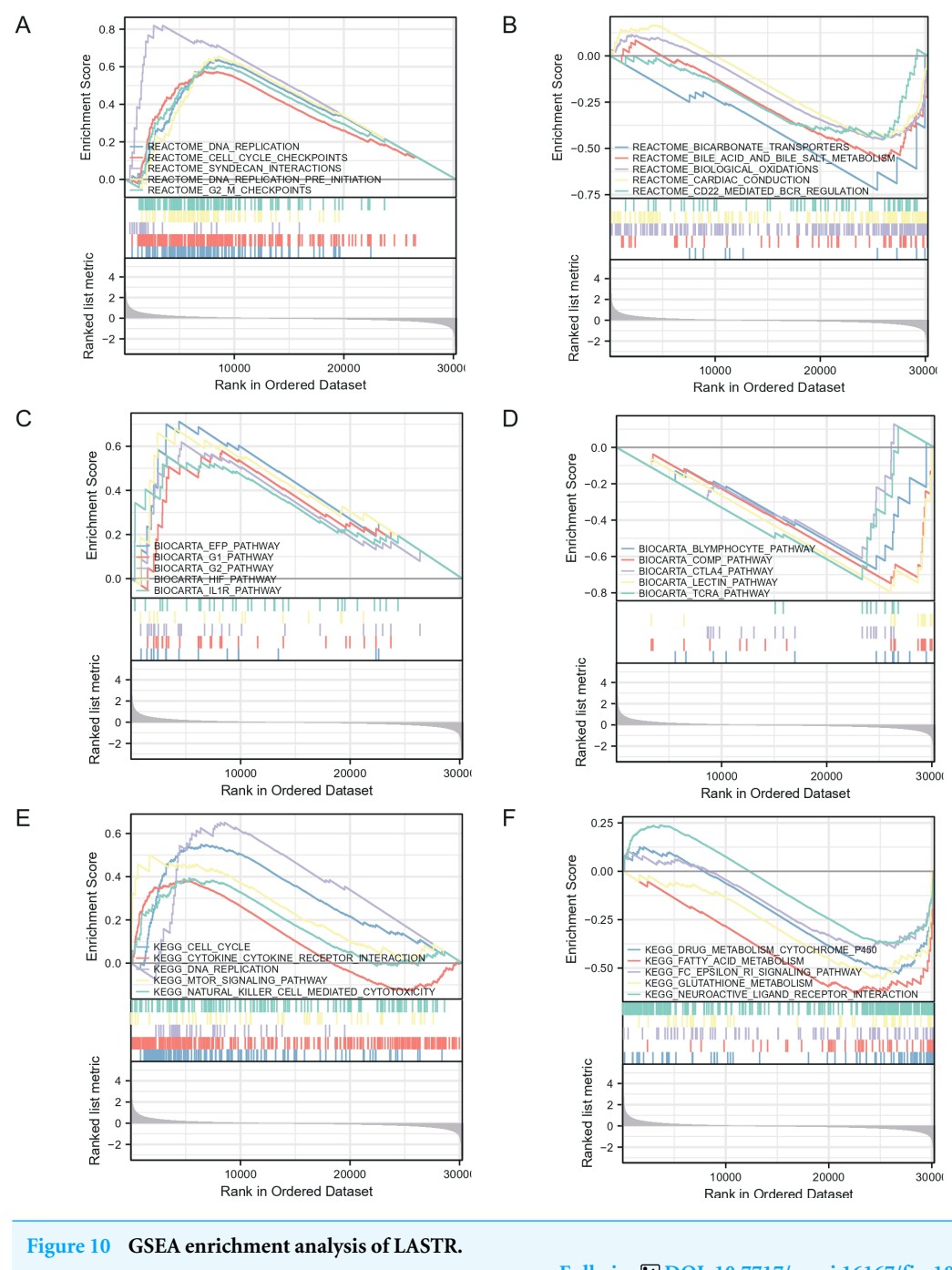

**Figure 10  GSEA enrichment analysis of LASTR.**

# DISCUSSION

In developing countries, lung cancer is the major cause of mortality from malignant tumors globally, and molecular aberrations of oncogenes are the main factors in the pathogenesis. However, the exact mechanism of LUAD pathogenesis remains unclear. Dysregulated splicing tends to be a prevalent event in cancers even where there are no mutations in the

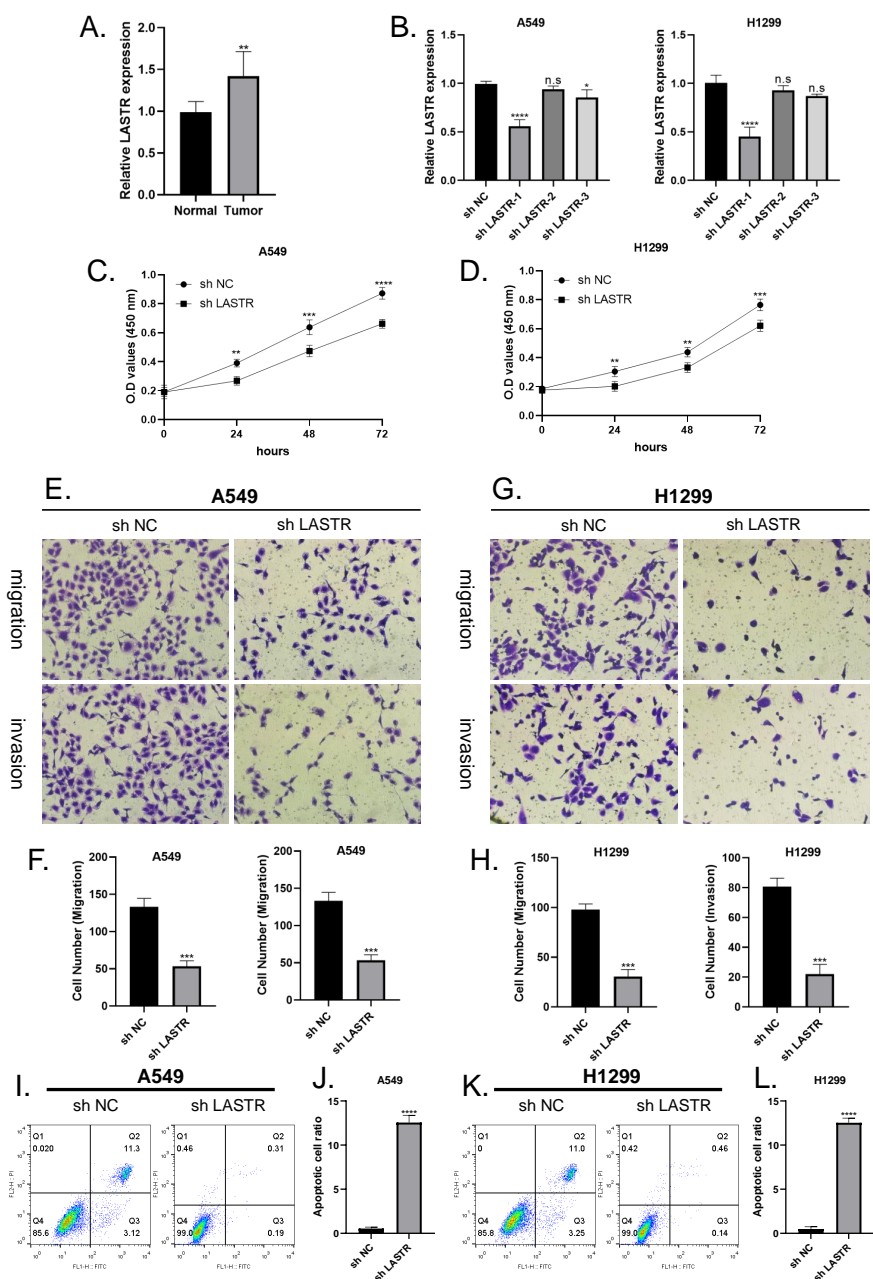

**Figure 11** **LASTR promotes lung cancer cell migratory and invasive capacity *in vitro*.** (A) Results of RT-PCR assay on tumor tissues and paraneoplastic tissues, LASTR was elevated in tumor tissues; (B) Detection of transfection efficiency of sh-LASTR in A549 cell line and H1299 cell line using RT-PCR, and standardization of the results. (C–D) CCK8 results after LASTR inhibition using shRNA in A549 and H1299 cell lines. (E, G) Transwell assay images of migration and invasion in the negative control and LASTR knockout groups. (F, H) Quantitative analysis of migrating and invading lung cancer cells. (I–J) Apoptotic flow cytometry results after interference with LASTR expression in A549 cell line. (K–L) Apoptotic flow cytometry results after interference with LASTR expression in H1299 cell line. n.s $P > 0.05$, $**P \leq 0.01$, $***P \leq 0.001$, $****P \leq 0.0001$.

core splicing machinery. As lncRNA expressions tend to be differentially modulated in various types of malignancy and their expression levels are linked to tumorigenesis, tumor aggressiveness, and stages, they might be targets for malignancy therapies. The existence of more modulating interaction sites in lncRNAs avails a broader platform for creating novel structure-based anti-cancer agents. Additionally, with their engagement in distinct cell signaling pathways as well as tissue-specific expression, lncRNAs can devise novel approaches for particular malignancy subtype diagnosis as well as targeting (*Ren et al., 2021*). Preceding their breakthrough in targeted lncRNA-based gene treatment, researchers screened innovative small-molecule libraries and conducted several clinical studies (*Boshier et al., 2018*).

Apart from playing a part in the evaluation of the direct impact of lncRNAs in neoplasm diagnosis and management, they can be additionally involved in enhancing therapeutic efficacy as well as the creation of combination therapy (*Lu et al., 2022*). Therapeutic resistance poses a great threat to the management of malignancies (*Cruz-Jentoft et al., 2010*). Nevertheless, this might be addressed by improving the therapeutic sensitivity of malignancies as a result of resistance by regulating critical cell signaling pathways (*Fulop et al., 2019*). Because lncRNAs are closely linked to numerous cell signaling processes, the regulation of their expression might be performed to enhance the therapeutic sensitivity of malignancies. LncRNAs can be used to boost tumor treatment sensitivity and may also be used as well in combination therapy.

LncRNAs are promising agents, especially in the diagnosis of malignancies as well as therapy (*Williams et al., 2019*). The discovery of a large number of lncRNAs, their extensive expressions in different types of malignancy, tumor specificity, and their stability in body fluids (plasma and urine) discovery avail a novel cornerstone for creating diagnosis as well as management therapies for cancer (*Han et al., 2020*; *Wang et al., 2020*). LncRNA expression might be utilized to anticipate the prognosis as well as individuals' outcomes. LncRNAs are principal modulators of chromatin dynamics in addition to gene regulation and are linked to diverse cell signaling pathways, and their expressions are affected by diverse factors such as nutrients, age, hormones, and sex (*Chen et al., 2015*; *Hadji et al., 2016*; *Alvarez-Dominguez et al., 2015*). The abnormal long non-coding transcriptome represents an unknown level of post-transcriptional control in malignancy. We discovered that LINC02657 or LASTR (lncRNA linked to SART3 control of splicing) (*Wang et al., 2020*), a stress-induced lncRNA, is required for cancer growth (*Han et al., 2020*). In LUAD individuals, it is essential to discover possible diagnostic and prognostic LncRNAs. However, its involvement in LUAD is unknown. The expression of LASTR in LUAD and its link to clinical characteristics and prognosis in LUAD individuals were examined in this work.

In this investigation, we retrieved LUAD level 3 RNA-Seq data and respective clinical data from the public database of TCGA to evaluate whether LASTR expression is linked to the prognosis of LUAD cases. The expression level of LASTR was substantially elevated in LUAD tissues and corresponding normal lung tissues ($p < 0.001$), and it was verified by the ROC curve that LASTR expression could be used as an important parameter to distinguish normal tissues from tumor tissues. Meanwhile, LASTR was remarkably

associated with differential infiltration of different immune cells. The quantification of immune cells was computed by the CIBERSORT algorithm, the infiltration differences of 22 immune cells were analyzed based on two groups, and the comparison between LASTR and differentially infiltrated immune cells was done. Cell infiltration analysis of the immune microenvironment of LUAD patients showed that the proportion of Mast cells, TFH, Tcm, T helper cells, iDCs, Eosinophils, DCs, and B cells were significantly reduced in the high-LASTR group in the TCGA-LUAD cohort while The infiltration level of NK CD56dim cells, Th2 cells, and Tgd were significantly increased. Logistic regression was utilized to ascertain the correlation between LASTR expression and clinical characteristics of LUAD individuals. Kaplan-Meier method, as well as the Cox regression method, detected the impact of LASTR expression level on OS, and the nomogram analyzed the link between LASTR gene expression and the risk of LUAD. Kaplan-Meier survival analysis affirmed that LUAD demonstrated a poor prognosis in terms of OS, PFI, and DSS compared with high-expression LASTR and low-expression LASTR ($p < 0.005$). Logistic regression identified high-LASTR expression independent predictors for early death. Univariate as well as multivariate Cox regression analyses demonstrated that LASTR correlated independently with OS, and high LASTR expression was an independent factor influencing OS (HR =1.666, CI [1.241–2.237], $p < 0.001$). T stage (T3 & T4 *vs.* T1 & T2) ($p = 0.046$), N stage (N1 & N2 & N3 *vs.* N0) ($p = 0.002$), pathologic stage (Stage III & Stage IV *vs.* Stage I & Stage II) ($p = 0.011$). The relationship between LASTR expression and LUAD risk is described in the nomogram. Subsequent survival-prognostic analysis results verified that high LASTR expression was linked to poorer survival status and poorer prognostic clinicopathological features of LUAD patients. Logistic regression ascertained the association between LASTR expression and clinical features in LUAD cases and showed that high expression of LASTR was substantially correlated with T stage (T3 & T4 *vs.* T1 & T2) ($p = 0.046$), N stage (N1 & N2 & N3 *vs.* N0) ($p = 0.002$), Pathologic stage (Stage III & Stage IV *vs.* Stage I & Stage II) ($p = 0.011$). Nevertheless, the high expression of LASTR was not remarkably linked to other clinical features. Multivariate Cox regression analysis depicted that LASTR ($p = 0.016$), primary outcome therapy ($p = 0.004$), and tumor status ($p < 0.001$) could be independent prognostic factors for LUAD. Next, we then looked at the link between LASTR expression levels and the prognosis in LUAD. Correlation analysis in subgroups of clinical variables suggested, high LASR expression in T stage (T3 & T4 *vs.* T1 & T2) ($p = 0.046$), N stage (N1 & N2 & N3 *vs.* N0) ($p = 0.002$), pathologic stage (Stage III & Stage IV *vs.* Stage I & Stage II) ($p = 0.011$) associated with poor prognosis. The results also supported an independent prognostic analysis GO and KEGG enrichment analysis for genes with significant differences between the two subgroups in LUAD samples, and GSEA for group differences in LASTR expression in the TCGA-LUAD dataset (GSEA). GO enrichment analysis showed that LASTR was related to keratinization, keratin filament, the structural composition of the cytoskeleton, serine-type endopeptidase activity, and other functions. KEGG results suggested numerous pathways associated with tumor-related and neuroactive ligand-receptor binding. GSEA found that patients with high LASTR expression were differentially enriched in a number of pathways related to tumorigenesis. Our study reveals an unexpected oncogenic function of LASTR and provides a new

perspective for its research in tumors, but experimental validation is currently lacking. In addition to promoting the malignant phenotype of tumors, LASTR also has a potential function in the modulation of the tumor microenvironment, particularly in LUAD. These findings may provide a possible biomarker for malignancy prognosis as well as therapy. In conclusion, our studies so far demonstrate that increased LASTR levels can predict overall survival prognosis in LUAD patients and are significantly enriched in multiple tumor- and disease-progression-related pathways. high LASTR expression is an independent predictor of poor prognosis in individuals suffering from LUAD and affects the tumor immune microenvironment. The present study shows that the LASTR copy number is elevated in LUAD, suggesting that this molecular target is present and abundant in LUAD patients.

### Funding
The authors received no funding for this work.

### Competing Interests
The authors declare there are no competing interests.

### Author Contributions
- Fanming Kong conceived and designed the experiments, performed the experiments, analyzed the data, prepared figures and/or tables, authored or reviewed drafts of the article, and approved the final draft.
- Xinyu Yang conceived and designed the experiments, performed the experiments, analyzed the data, prepared figures and/or tables, authored or reviewed drafts of the article, and approved the final draft.
- Zhichao Lu conceived and designed the experiments, performed the experiments, prepared figures and/or tables, authored or reviewed drafts of the article, and approved the final draft.
- Zongheng Liu conceived and designed the experiments, prepared figures and/or tables, authored or reviewed drafts of the article, and approved the final draft.
- Yang Yang conceived and designed the experiments, analyzed the data, prepared figures and/or tables, authored or reviewed drafts of the article, and approved the final draft.
- Ziheng Wang conceived and designed the experiments, analyzed the data, prepared figures and/or tables, authored or reviewed drafts of the article, and approved the final draft.

### Data Availability
The original data can be extracted from the TCGA LUAD queue in the database of the gene expression spectrum (https://portal.gdc.cancer.gov/exploration).

## Supplemental Information

Supplemental information for this article can be found online at http://dx.doi.org/10.7717/peerj.16167#supplemental-information.

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
