# Peer review of "A novel long noncoding RNA (lncRNA), LINC02657(LASTR), is a prognostic biomarker associated with immune infiltrates of lung adenocarcinoma based on unsupervised cluster analysis"

_PeerJ, doi:10.7717/peerj.16167_

## Round 0.1 · original submission · Major Revisions

Please address the Reviewer's comments carefully.

Reviewer 1 ·

Basic reporting

Thank you for manuscript me the article entitled " A novel Long Noncoding RNA (lncRNA), LINC02657(LASTR), is prognostic biomarker associated with immune infiltrates of Lung Adenocarcinoma " for evaluation. This research is aimed at the bioinformatics data mining of lung adenocarcinoma (LUAD). Based on the bioinformatics data mining of the TCGA public database, suggested that LASTR is a potential molecular biomarker that can predict the prognosis and tumor immune microenvironment of LUAD patients. At the same time, the expression of LASTR was excavated to correlate with the infiltration level of various immune-related cells in LUAD. Through immune correlation analysis, the important role of tumor microenvironment (TME) in the occurrence and progression of LUAD was explored.
1. This is a bioinformatic analysis using data sets from the Cancer Genome Atlas. It is a cluster analysis based on the grouping of LASTR expression, exploring a set of pivotal genes related to the clinical prognosis of LUAD and tumor immune infiltration. However, the authors could not explain the relation with these biomarkers and LUAD.
2. An interesting manuscript in its field, I suggest that the authors add a comment how they will use this novel biomarker in the everyday clinical practice.
3. Although the article explored and verified the effect of LASTR on the clinicopathological characteristics and overall survival prognosis of LUAD patients through a variety of ways, there is a lack of correlation studies with these clinical characteristics and needs to be supplemented.
4. A summary paragraph at the beginning telling the reader what the results of this study were, and whether this supports the hypotheses or not, is needed.
5. There are terms such as immune microenvironment and immune-related tumor microenvironment in the article. Please consider whether these concepts are the same. The article should correctly describe these special pronouns and unify their meaning in research.
6. There are some grammatical errors in the article, which need to be checked to correct the language errors in the article.
The article involved GO and KEGG enrichment analysis, but did not show significant differences in specific enrichment results, please supplement in the text.
It is important to consider caveats or alternative interpretations of this work.

Overall the manuscript can be scientifically evaluated again after a major revision.

Experimental design

Wll

Validity of the findings

Well

Additional comments

N/A

·

Basic reporting

The manuscript is novel and compelling, although there are some problems in presenting the results. It would benefit from proofreading and editing. There were some difficult phrasings, grammar issues, and misspellings. The title is also descriptive of the content, which cannot summarize the innovation of the article well. For the incomplete description of how to cluster in the method, it is necessary to clarify the principle of consistent clustering. Although the paper showed evidence of LASTR has a certain correlation with the immune microenvironment, the discussion part has not been able to clarify its potential application value. Please describe them appropriately. The article involved GO and KEGG enrichment analysis but did not show significant differences in specific enrichment results, please supplement the text. The most important point is that the article only conducts multi-level data mining analysis on the molecular expression data of TCGA database.
Please consider editing the paper so that each paragraph has a topic and concluding statement, with support for these in between. Overall the manuscript can be scientifically evaluated again after a major revision.

Experimental design

Although the content is relatively substantial, there is still a lack of experiments to verify the expression difference of LASTR in normal and lung cancer patients. If the experimental data can be supplemented, the research conclusions can be more convincing. Finally, some more wet experiments should be added.

Validity of the findings

No comment.

Additional comments

No more additional comments.

---

## Round 0.2 · accepted · Accept

The authors have well answered the comments from Editors and Reviewers.

Reviewer 1 ·

Basic reporting

No more comments.

Experimental design

No more comments.

Validity of the findings

No more comments.

Additional comments

No more comments.

·

Basic reporting

accept

Experimental design

accept

Validity of the findings

accept

Additional comments

accept